# Self-Assembly Preparation of Nano-Lignin/Cationic Polyacrylamide Complexes

**DOI:** 10.3390/polym13111726

**Published:** 2021-05-25

**Authors:** Guoyu Tian, Xiuhong Zhong, Xuehai Wu, Zhaojiang Wang

**Affiliations:** 1Key Laboratory of Green Printing & Packaging Materials and Technology in Universities of Shandong, School of Light Industry Science and Engineering, Qilu University of Technology (Shandong Academy of Sciences), Jinan 250353, China; tianguoyutgy@163.com; 2Key Laboratory of Pulp and Paper Science & Technology of Ministry of Education, Qilu University of Technology (Shandong Academy of Sciences), Jinan 250353, China; zhongxiuhong1223@163.com (X.Z.); wxh372835@163.com (X.W.)

**Keywords:** nano-lignin, self-assembly, ethanol–water system, fluorescence intensity

## Abstract

The present work describes the preparation of nano-lignin particles from calcium lignosulfonate (CL). The nano-lignin was fabricated from colloidal lignin–polyacrylamide complexes via self-assembly. The sizes of the nano-lignin particles were examined by dynamic light scattering (DLS) and scanning electron microscopy (SEM). The results indicated that the average particle size of the prepared nano-lignin was approximately 100 nm. In addition, the obtained nano-lignin exhibited enhanced fluorescence intensity when compared with the original lignin, which might represent a potential application of this nano-particle product.

## 1. Introduction

Lignin is the world’s most abundant renewable and biodegradable natural resource, after cellulose [1]. It is an amorphous polymer formed from the linking of various phenylpropane units through ether and carbon–carbon bonds [2]. Lignin is composed of the following three types of phenylpropane units: p-hydroxyphenyl (H), guaiacyl (G), and syringyl (S) [3]. As a side product from the pulping industry, lignin has an annual output of 70 million tons, 95% of which is used as fuel or discharged with pulp black liquor; only 5% is used to produce high-value added products, including additives, dispersants, adhesives, and surfactants [4]. In most cases, this spent liquor is burned to recover the spent inorganic chemicals, whereby the dissolved lignin is an additional source of organic fuel. The collection of lignin from waste liquid not only serves to mitigate the pollution caused by the pulping industry, but it also plays an important role in the reduction in mineral resources, contributing to the sustainable development of the social economy [5,6].

Nanoscience technology is quickly becoming one of the main drivers of economic development. Existing nanomaterials are mostly derived from non-degradable petroleum and mineral products. However, renewable feedstocks from biomass are receiving increased attention as alternative feedstocks to address the energy and environmental issues associated with non-renewable resources. The development of nano-lignin can upgrade this bioresource to a high-value product [7]. Nano-lignin particles have a higher specific surface area when compared with micron- and larger-sized lignin particles. When mixed with various polymers, these nanoparticle adjuncts closely interact with, and are evenly dispersed throughout, the polymer matrix. Lignin nanoparticles can improve the thermal stability and mechanical and barrier properties of the resulting composite material [8]. In addition, some studies have indicated that nano-lignin particles exhibit antibacterial and non-cytotoxic properties [9,10,11].

At present, nano-lignin is synthesized by various methods, including electrospinning [12], sol-gel [13], and supercritical solvent extraction [14]. In addition, enzymatic hydrolysis combined with physical pulverization [15], ultrasonication [16], in-situ composite formation [17], ionic liquid treatment [18], and other novel methods [19] have been used to generate nano-lignin. The sol-gel method uses mild reaction conditions, but the sol condensation process can take several days to several weeks to occur; often, the prepared nano-materials are not uniformly distributed, and the product yields are very low. The nano-lignin prepared by the supercritical extraction method yields small-sized particles that have a narrow size distribution with little particle agglomeration. The disadvantage of this method is its high capital cost for pressurized equipment, which makes the process industrially unviable [20].

In this paper, nano-lignin was efficiently prepared by electrostatic self-assembly and self-aggregation. The resulting nano-lignin exhibits higher fluorescence, which may make this material useful for developing fluorescent materials.

## 2. Materials and Methods

### 2.1. Materials

Calcium lignosulfonate (CL) was purchased commercially (Georgia-Pacific Corporation, Augusta, GA, USA). It is a powder obtained by sulfonation of softwood lignin. The basic properties of CL were provided by Georgia-Pacific Corporation as detailed in Table 1 [21]. Cationic polyacrylamide (CPAM) with a molecular weight ranging from 8 million to 18 million Da and ionization degree of 25% was obtained commercially (Shenghuang Chemical Products Co., Ltd., Liaocheng, Shandong, China).

### 2.2. Nano-Lignin Preparation

CPAM was slowly added to deionized water at room temperature with vigorous stirring to prepare a cationic polymer electrolyte solution (1000 mg/L). CL was dissolved in deionized water to obtain an anionic polymer electrolyte solution (2000 mg/L). While vigorously stirring the cationic solution at room temperature, the anionic solution was slowly added dropwise to the cationic solution at a CL/CPAM mass ratio of 100:1. This is an optimized ratio by reaching a zeta potential of −12 mV to stabilize nano-particles. At the fixed concentration of CPAM, flocculation occurs when CL:CPAM is bigger than 100:1 due to formation of CL–CPAM precipitate. Otherwise, the self-assembly is time-consuming when the CL:CPAM is less than 100:1, which means poor efficiency preparation of nano-lignin. This mixing process resulted in the formation of nano-lignin particles. The concentration dependence of the in-situ nano-lignin formation was studied over a wide range of CL solution concentrations, from 100 to 1000 mg/L.

### 2.3. Characterization

The zeta potential and the size of nano-lignin particles were measured using a Malvern Nano ZS90 instrument (Malvern Panalytical Ltd.; Royston, UK) that operated at 25 °C. Particle size was measured from 0.3 nm to 5 μm (diameter) using a 90-degree scattering optical element with a tunable laser (power: 50 mW; wavelength: 532 nm). All measurements were performed in triplicate. The optical properties of the nano-lignin solutions were characterized using an Agilent 8453 UV–Vis spectrophotometer (Agilent; Santa Clara, CA, USA) and a Hitachi F-4500 fluorescence spectrophotometer (Hitachi; Tokyo, Japan). SEM images of the lyophilized (freeze-dried) nano-lignin was acquired using a Hitachi Regulus 8220 scanning electron microscope (Hitachi; Tokyo, Japan) that operated at an acceleration voltage of 5 kV. A thin layer of gold was sputtered onto the surfaces of the samples prior to imaging using a BSC-ETD2000 sputter coater (Columbia International, Irmo, SC, USA).

## 3. Results and Discussion

### 3.1. Self-Assembly Proocess

Anionic and cationic polymers can form complexes with one another over a wide range of stoichiometric mass ratios. These complexes tend to be water-soluble unless there is charge neutrality in the applied ratios of the two ionic polymers. When dissolved in water, CL exhibits a negative electrical charge. The surface charges were measured as expressed by zeta-potential. The zeta-potential of CL showed a wide range from −30 mV to −20 mV depending on pH at 25 °C, which is consistent with references [22]. The ionization of sulfonic acid groups and phenolic hydroxyl groups at higher pH values generates more negative charges and resulted in a much lower zeta potential. Oppositely, cationic polyacrylamide (CPAM) possesses positive charges and has a positive zeta potential from 7–12 mV depending on pH value. Colloidal CL is prepared by adding CPAM solution. Excess CPAM solution will cause the CL colloid to flocculate prematurely and reduce the amount of CL adsorbed. Stable nano-lignin particles were obtained with an optimum mass ratio of 100:1 of CL:CPAM. The CL:CPAM of 100:1 resulted in a zeta potential value of −13 mV, negative enough to stabilize the new formed nano-lignin particles. Figure 1 illustrates the mechanism of nano-lignin formation when CPAM interacts with CL.

### 3.2. Dynamic Groeth of Nano-Lignin

As mentioned earlier, an optimum mass ratio of 100:1 CL-to-CPAM was selected for nano-lignin preparation. A high solution concentration of CL is preferred to maximize the production efficiency of nano-lignin. In this regard, a wide range of CL concentrations, from 100 to 1000 mg/L, was investigated. The solvents used in this study were all ultra-pure water, and the reaction system was at natural pH around pH 7.

As shown in Figure 2a, the size of nano-lignin at different CL concentrations was measured. The size of the nano-lignin particles was dependent on the CL concentration, increasing from 100 nm at 112 mg/L CL to 600 nm at 1000 mg/L CL.

Figure 2b shows the dynamic growth of nano-lignin particles at various concentrations of CL. The particle size increased exponentially with the passage of time. Furthermore, the particle growth rate is positively correlated with the CL concentration. To obtain nano-lignin that is relatively small in size, it is necessary to quench the kinetic particle growth by lyophilization. Figure 2c shows the particle size distribution of the nano-lignin after 20 min. The distributions tended to be wider when the CL concentration was higher, which confirms the self-assembly process using cationic and anionic polymer electrolytes.

### 3.3. Topography Analysis

It is very difficult to obtain nano-sized lignin particles because CL/CPAM complexes grow exponentially in a relatively short time period (30 min). This exponential growth of the complexes leads to spontaneous coagulation when particles become too large to maintain a stable sol. The resulting particles were lyophilized to stop their growth and were then subjected to SEM analysis to characterize their surface topography. Figure 3a shows CL/CPAM particles with sizes of approximately 5 to 60 µm. The big CL/CPAM particles from lyophilization with a size larger than 10 µm in Figure 3a proved the coagulation. This observation indicates that lignin aggregation inevitably occurs by the co-precipitation process in the process of lyophilization. Fortunately, there are numerous small particles with sizes less than 5 µm in Figure 3a. The magnified micrograph of the CL/CPAM particles (Figure 3b) clearly shows that they were of an oblate spheroid shape. There were a lot of tiny particles on the surfaces of the larger spheroids, which suggests layer-by-layer assembly of CL/CPAM complexes.

### 3.4. Nano-Lignin Fluorescence Intensity Analysis

Fluorescence is the emission of light by a substance that has initially absorbed electromagnetic radiation at a shorter wavelength. The excited atom or molecule emits radiation of the same or lower wavelength as the excitation radiation during de-excitation. When the excitation source stops irradiating the sample, the re-emission process immediately ceases. The re-emitted light is called fluorescence.

It was observed from the ultraviolet–visible (UV–Vis) absorption spectra for various CL concentrations (Figure 4) that the largest absorption peak occurred at 210 nm. The intense absorption at 210 nm is ascribed to the π−π* transition in benzene rings of CL. Actually, CL is modified chemically from lignin by sulfonation reactions. There are plenty of unsaturated structures in lignin, such as benzene ring, C=C, and C=O [23]. The CL excitation wavelength for the fluorescence spectrum also occurred at 210 nm. Figure 5 shows the fluorescence spectra for various CL concentrations at an excitation wavelength of 319 nm. From the spectra, the fluorescence intensity increased as the CL concentration increased until 600 mg/L. When the CL concentration reached 600 mg/L, the fluorescence intensity peaked at 215 a.u. As the CL concentration increased above 600 mg/L, the fluorescence intensity decreased. Figure 6 shows the peak fluorescence intensity of nano-lignin at various CL concentrations. As the CL concentration increased to 400 mg/L, the fluorescence intensity increased. When the CL concentration reached 400 mg/L, the fluorescence intensity was at its highest level; this intensity gradually decreased as the concentration increased to 600 mg/L.

The prepared samples exhibited aggregation-induced emission (AIE), where the lignosulfonate displayed intrinsic aggregation behavior. The AIE mechanism is involved in limiting intramolecular motion, including restricted intramolecular rotation (RIR) and restricted intramolecular vibration (RIV), as well as carbonyl groups. Both 1,1,2,2-Tetraphenylethylene (TPE) and hexaphenylsilole (HPS) are well-known and well-studied AIE luminogens. The fluorescence emissions of these compounds are enhanced when their concentrations are high in solution or when they exist in a solid state; their fluorescence behavior is governed by the RIR mechanism (Fan et al. 2008; Lam et al. 2008). Similarly, as the concentration of lignosulfonate increased, the particle sizes increased and the fluorescence emissions initially increased. When the CL concentration reached 800 mg/L, the fluorescence intensity started to decrease, which was caused by the AIE behavior of the lignosulfonate.

## 4. Conclusions

Nano-lignin was prepared via a self-assembly method using calcium lignosulfonate (CL) and cationic polyacrylamide (CPAM). Dynamic light scattering analysis was used to measure the particle sizes of nano-lignin in the aqueous phase. The CL/CPAM complexes were observed to grow exponentially over a short period of time (30 min). The solid CL/CPAM complexes obtained from lyophilization were oblate spheroid in shape.A mass ratio of 100:1 CL-to-CPAM was used for nano-lignin preparation. A high concentration of CL is preferred to maximize the production efficiency of nano-lignin.The sizes of the nano-lignin particles exponentially increased over a 30-min time interval; this growth rate is positively correlated with the CL concentration. To obtain nano-lignin with relatively small sizes, it is necessary to quench the growth by lyophilization.The fluorescence emission intensity increased as the concentration of CL increased. When the CL concentration was 600 mg/L, the fluorescence intensity was 215 a.u.; when the CL concentration was higher than 600 mg/L, the fluorescence emission intensity decreased. When the CL concentration was at 400 mg/L, the fluorescence emission intensity was at its maximum value; this emission intensity gradually decreased as the CL concentration increased from 400 to 600 mg/L.

## Figures and Tables

**Figure 1 polymers-13-01726-f001:**
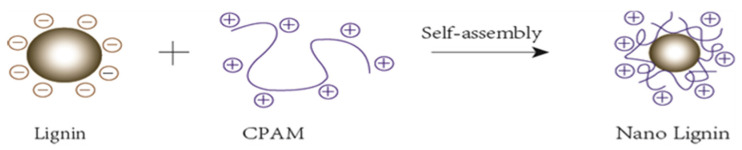
Schematic of the dynamic growth of nano-lignin when CL (lignin) is added to CPAM.

**Figure 2 polymers-13-01726-f002:**
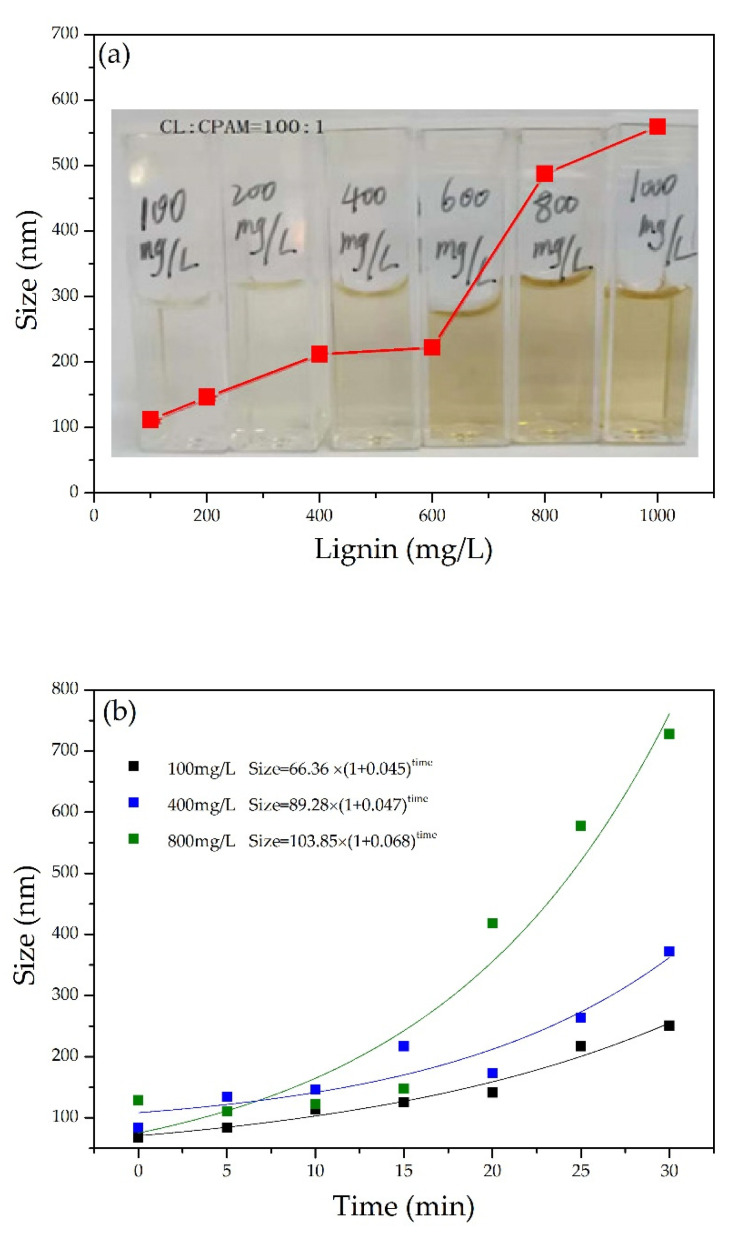
(**a**) Nano-lignin particle size at various CL concentrations; (**b**) dynamic growth of nano-lignin particles at various CL concentrations; and (**c**) size distribution of nano-lignin obtained at various CL concentrations. All self-assembly experiments were conducted with the fixed C_CL_:C_CPAM_ of 100:1 at room temperature and natural pH using pure water as buffer.

**Figure 3 polymers-13-01726-f003:**
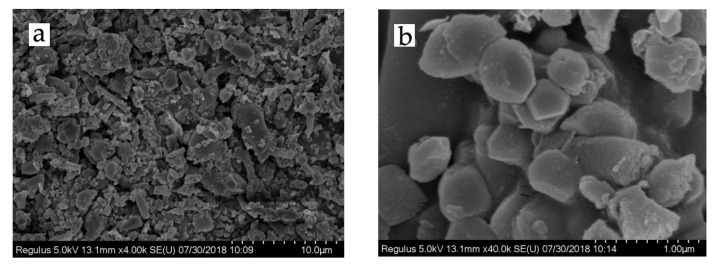
SEM images of CL/CPAM complexes at a CL concentration of 100 mg/L and a CL-to-CPAM ratio of 100:1 with (**a**) 4000 times magnification and (**b**) 40,000 times magnification.

**Figure 4 polymers-13-01726-f004:**
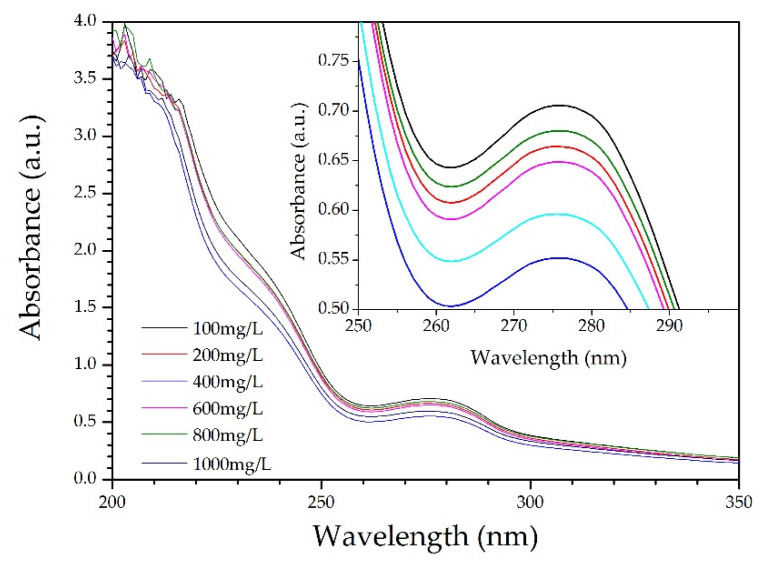
UV–Vis absorption spectra for various CL concentrations at 7 pH.

**Figure 5 polymers-13-01726-f005:**
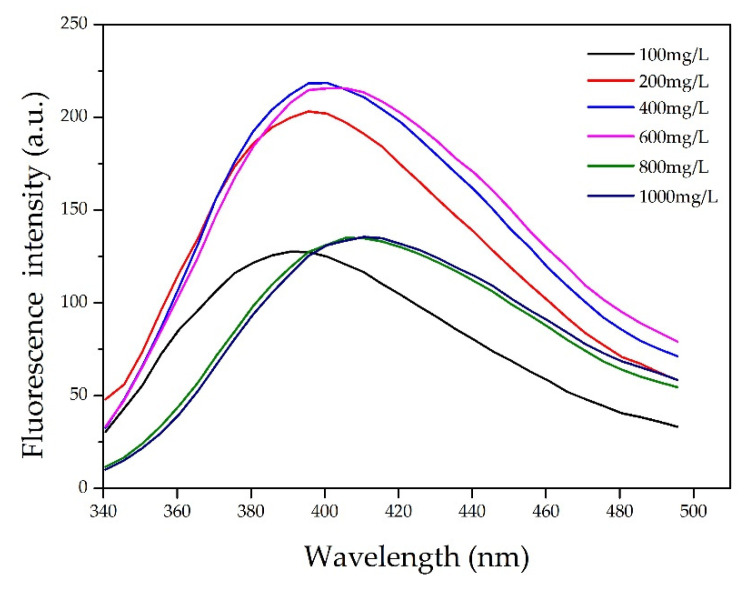
Fluorescence spectra of nano-lignin at various CL concentrations when excited by electromagnetic radiation at 319 nm.

**Figure 6 polymers-13-01726-f006:**
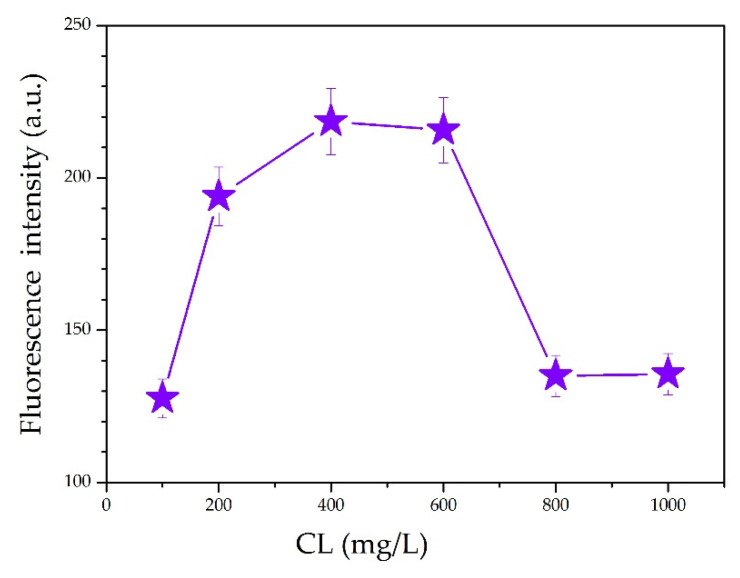
The strongest fluorescence intensity of nano-lignin at various CL concentrations.

**Table 1 polymers-13-01726-t001:** Properties of calcium lignosulfonate.

Mw (g/mol)	18,000
Mn (g/mol)	2500
pH	2.0–4.0
Insoluble (%)	1.5 max
Reducing substances (%)	11 max
Moisture (t%)	8 max
Molecular weight by GPC (g/mol)	19,200
Calcium (%)	4.0–5.5
Sodium (%)	0.02

## Data Availability

The data presented in this study are available on request from the corresponding author.

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
