# Peer review of "Self-Assembly Preparation of Nano-Lignin/Cationic Polyacrylamide Complexes"

_polymers, 2021, doi:10.3390/polym13111726_

Round 1

Reviewer 1 Report

In this manuscript, Tian et al. have described the fabrication of lignin nanoparticles from lignosulfonate. The authors have characterized the particle systems with the help of DLS and Microscopy tools. Though the research field of lignin-based nanoparticles is gaining much interest to find the potential applications of this one of the most abundant building block of nature, this study in my opinion does not bring any significant advancement in the field. It might still be of interest to the readers. I would suggest that authors should do the major changes in the manuscript as listed below, before it is being considered for publication:

  1. Authors must characterize the starting lignin. e.g. SEC, NMR and hydroxyl content. Also, HSQC for inter unit linkages and PNMR for hydroxyl content.
  2. Figure 2 is not very clear. Authors should have double axis to normalize.
  3. How doable is this process in industrial scale? The freeze drying process to stop the growth is definitely not optimal. Have the authors thought to fractionate lignin instead in terms of molecular weight?
  4. Is the lignin used in this study derived from softwood or hardwood ?
  5. Authors should investigate kraft lignin as well and compare the results. 

Reviewer 2 Report

In this work, the nanoparticles were obtained by self-assembly using calcium lignosulfonate (CL) and cationic polyacrylamide (CPAM). The investigation of these particles is interesting, but the quality of design and presentation of manuscript is very poor. Moreover, I also have the following comments:
1. Based on what experimental data was taken the optimal mass ratio of 100:1 CL to CPAM for stable nanolignin particles? What determines this stability? Why can't the 1:100 ratio CL to CPAM be conducive to the formation of stable particles?
2. Why was only one concentration of CPAM (1000 mg/L) used when mixed with different amounts of CL? In the caption of Figure 2, it is necessary to indicate the concentration of CPAM and the pH of the solutions.
3. The experimental section Materials and Methods states that the zeta potential values were measured, but no zeta potential data is available in the Results and Discussion. Please provide the dependence of the zeta potential on the ratio of CL and CPAM to confirm the electrostatic mechanism of mixed aggregation.
4. What electronic transition is responsible for absorption at 210 nm?
5. Figure 4 shows the spectra at pH 7. Is it a natural pH or a buffer?
6. The authors have carelessly formatted the article. The article contains words with a capital letter in the middle of sentences and paragraphs consisting of one and two sentences. Authors need to provide more discussion of their results and structure the whole manuscript.

Round 2

Reviewer 1 Report

Authors have implemented the comments from the reviewer. I, therefore, think that the manuscript is ready for publication now. 

Author Response

Thank you for your comments and suggestions on the manuscript we submitted. Best wishes.